# Medical Expulsive Therapy for Pediatric Ureteral Stones: A Meta-Analysis of Randomized Clinical Trials

**DOI:** 10.3390/jcm12041410

**Published:** 2023-02-10

**Authors:** Pardis Ziaeefar, Abbas Basiri, Moein Zangiabadian, Jean de la Rosette, Homayoun Zargar, Maryam Taheri, Amir H. Kashi

**Affiliations:** 1Urology and Nephrology Research Center (UNRC), Shahid Labbafinajad Hospital, Shahid Beheshti University of Medical Sciences (SBMU), Tehran 15167-45811, Iran; 2Department of Urology, Istanbul Medipol Mega University Hospital, Istanbul 34214, Türkiye; 3Department of Urology, Royal Melbourne Hospital, University of Melbourne, Melbourne 3050, Australia; 4Department of Surgery, University of Melbourne, Melbourne 3010, Australia

**Keywords:** pediatrics, meta-analysis, randomized controlled trial, ureteral stones, medical expulsive therapy, adrenergic alpha-antagonists

## Abstract

To evaluate the efficacy and safety of medical expulsive therapy (MET) for ureteral stones in pediatric patients, Cochrane, PubMed, Web of Science, Scopus, and the reference list of retrieved studies were searched up to September 2022 to identify RCTs on the efficacy of MET. The protocol was prospectively registered at PROSPERO (CRD42022339093). Articles were reviewed, data were extracted by two reviewers, and the differences were resolved by the third reviewer. The risk of bias was assessed using the RoB2. The outcomes, including the stone expulsion rate (SER), stone expulsion time (SET), episode of pain, analgesic consumption, and adverse effects, were evaluated. Six RCTs enrolling 415 patients were included in the meta-analysis. The duration of MET ranged from 19 to 28 days. The investigated medications included tamsulosin, silodosin, and doxazosin. The stone-free rate after 4 weeks in the MET group was 1.42 times that of the control group (RR: 1.42; 95% CI: 1.26–1.61, *p* < 0.001). The stone expulsion time also decreased by an average of 5.18 days (95% CI: −8.46/−1.89, *p* = 0.002). Adverse effects were more commonly observed in the MET group (RR: 2.18; 95% CI: 1.28–3.69, *p* = 0.004). The subgroup analysis evaluating the influence of the type of medication, the stone size, and the age of patients failed to reveal any impact of the aforementioned factors on the stone expulsion rate or stone expulsion time. Alpha-blockers as medical expulsive therapy among pediatric patients are efficient and safe. They increase the stone expulsion rate and decrease the stone expulsion time; however, this included a higher rate of adverse effects, which include headache, dizziness, or nasal congestion.

## 1. Introduction

Urolithiasis is a worldwide problem, and its prevalence is estimated at roughly 10% in men and 7% in women in the United States [1,2]. Urolithiasis incidence among children accounts for 2–3% of total cases [3] and includes 1/1000 to 1/7600 of pediatric hospitalizations in the USA [4]. The prevalence of ureteral stones in countries with a high quality of life is over 10% [5].

The management of ureteral stones in children includes open surgery, ureteroscopy (URS), extracorporeal shock-wave lithotripsy (ESWL), and percutaneous nephrolithotomy (PCNL). However, these procedures are costly and invasive and require anesthesia [6]. In recent years, MET (medical expulsive therapy) including alpha antagonists, steroids, and calcium channel blockers, has been used to enhance the distal ureteral stone expulsion rate in adults [6]. Regarding the high risk of stone recurrence in patients, invasive treatments need repetition, and successful observations will be associated with less of a risk for urinary system manipulation and anesthesia. The main treatment strategy for the treatment of pediatric ureteral stones in many centers has been observation through MET, with a 65% higher stone expulsion rate compared to that of a placebo. [7] Nevertheless, studies on MET in children are limited. The smaller ureter of children raises concerns regarding the success of MET. The quality of MET studies in children is low, and their results are inconsistent. Two meta-analyses have been published on MET in children [8,9], with the latest one including 212 pediatric patients. Considering the significant number of studied patients after the latest meta-analysis, we decided to prepare an updated meta-analysis to explore the effect of MET and its contributing factors and strengthen the power of its conclusions.

## 2. Materials and Methods

### 2.1. Search Strategy

A systematic review was performed in June 2022 based on searching online databases, including Cochrane Central Register of Controlled Trials, PubMed, Web of Science, and Scopus, for articles published after January 2000. The following search terms were used: (medical expulsive therapy OR ureteral stone) AND (child OR children). We developed and examined a population, intervention, comparison, and outcome (PICO) question following Cochrane procedures: (P)atients include pediatric patients aged 18 or younger with distal ureter stones sized less than 12 mm; the (I)ntervention is using medical expulsive therapy with one of the alpha-adrenergic blocker drugs or other drugs such as calcium channel blockers, corticosteroids or other adjunctive medications; the (C)omparison group included patients who received analgesic medications and/or placebo; the (O)utcomes would be the determination of the SER (stone expulsion rate), the SET (stone expulsion time), the episode of pain, the analgesic consumption, or the adverse effects. The online database searching was started in June 2022 and was updated in September 2022 to investigate newly published articles. The protocol was prospectively registered at PROSPERO (CRD42022339093).

### 2.2. Study Selection

The article selection was conducted according to the search strategy recommended in the Preferred Reporting Items for Systematic Reviews and Meta-analyses criteria (PRISMA Figure 1). The search and selection criteria were not limited to the English language. Randomized controlled trials (RCTs) that met the following criteria were included; full-text-available study on the efficacy of medical expulsive therapy on the ureteral stone passage of pediatrics, the study provided accurate data of a minimum sample size of 10 cases, and the study included patients below 18 years old with distal ureteral stones ≤12 mm. Studies that did not satisfy the aforementioned criteria were excluded from the analysis. The articles found through the database searching were merged, and the duplicates were removed. Two reviewers (P.Z. and M.T.) independently screened the records by the title/abstract and the full text to exclude those irrelevant to the study topic. Studies that met the following criteria were included: (1) patients including pediatrics with ureter stones; (2) patients who used medical expulsive therapy; and (3) reported outcomes (SER, SET, episode of pain, analgesic consumption, or adverse effects). Reviews, conference abstracts, letters, correspondences, case reports/series, and studies that were not performed in the format of a randomized clinical trial were excluded. 

Two reviewers (P.Z. and M.T.) independently reviewed and included articles from the searches during the title/abstract screening; then, they examined the full texts of the indicated articles. Differences were resolved by consulting a third reviewer (A.K). The reference lists of the retrieved studies were also explored to identify RCTs on the efficacy of medical expulsive therapy in children.

### 2.3. Data Extraction

The following data were extracted from each study and placed into an excel spreadsheet (Microsoft, Redmond, WA): first author’s name; year of publication; the number of pediatric patients with ureter stones; the separated number of patients based on their gender; patient age; stone size; interventions; study duration; and treatment outcomes (including the SER, SET, episode of pain, analgesic consumption, or adverse effects). We emailed the authors of the selected articles to request that they provide us with further data regarding the stone expulsion rate and stone expulsion time based on different categories of age (≤2, 3–6, ≥7 years), stone size (<5 mm, 5–10 mm, > 0 mm), gender (boy/girl), and side (left/right).

### 2.4. Risk of Bias Assessment

The RoB2 recommended by the Cochrane Collaboration was used to assess the quality of the RCTs included in this study. Two reviewers (P.Z. and A.K.) independently assessed studies in five domains. Any disagreement was settled by discussion. 

### 2.5. Statistical Methods

The pooled risk ratios (RRs) with a 95% confidence interval (CI) for dichotomous data (adverse effects and stone expulsion rate) and the pooled difference in means with a 95% confidence interval (CI) for continuous data (stone expulsion time) were assessed by using random or fixed-effect models. The between-study heterogeneity was assessed by Cochran’s Q and the I2 statistic. I2 values more than 50% were considered as high heterogeneity [10]. The fixed-effects model was used when the estimated heterogeneity of the true effect sizes was low, and the random-effects model was used in high-heterogeneity studies. Moreover, subgroup analysis was performed to compare the effect of different treatment regimens, stone sizes, and ages on each outcome. Publication bias was evaluated statistically by using Egger’s and Begg’s tests as well as the funnel plot (*p* < 0.05 was considered indicative of statistically significant publication bias; funnel plot asymmetry also suggests bias) [11]. All analyses were performed using “Comprehensive Meta-Analysis” software, Version 2.0 (Biostat, Englewood, NJ, USA).

## 3. Results

### 3.1. Search Result

The literature search yielded a total of 262 studies. Of these, 26 studies were selected for full-text screening; ultimately, 6 studies met all the inclusion criteria and were included in this meta-analysis. The study selection process is illustrated in Figure 1. The baseline, clinical, and treatment-related characteristics of all the included studies are demonstrated in Table 1. The RCT studies included in this study were conducted in two countries: Egypt and Turkey. A total of 415 patients were included in this study. Among the 415 patients, 251 were boys and 164 were girls. The duration of intervention in the selected articles ranged from 19 days to 4 weeks. All the studies used alpha-adrenergic antagonists including tamsulosin, silodosin, and doxazosin [3,12,13,14,15,16].

### 3.2. Risk of Bias

Three studies [3,13,14] had a low risk of bias in all five domains (Figure 2). The study conducted by Aydogdu et al. [12] had a high risk of bias regarding the randomization process, and no information was mentioned regarding deviations from the intended intervention. Concerns regarding missing data were mentioned in two studies [15,16].

### 3.3. Treatment Outcomes

The present meta-analysis indicated that medical treatment in children with ureteral stones could significantly increase the stone expulsion rate (RR: 1.42; 95% CI: 1.26–1.61, *p* < 0.001, I2: 4.42, Figure 3) and also significantly decrease the stone expulsion time (difference in means: −5.18; 95% CI: −8.46/−1.89, *p* = 0.002, I2: 98.4; Figure 4). However, adverse effects in the treatment group were significantly more common than they were in the control group (RR: 2.18; 95% CI: 1.28–3.69, *p* = 0.004, I2: 0.0; Figure 5). There was no evidence of publication bias in any of the investigated outcomes, and the corresponding *p*-values were not statistically significant. 

### 3.4. Subgroup Analysis

The effects of different treatment regimens, stone sizes, and ages on the SER and SET are shown in Table 2 and Table 3. Both Tamsulosin and Silodosin had a statistically significant effect on SER increases and SET decreases, but the effect of Doxazosin on these outcomes was not statistically significant. Nevertheless, the average risk ratio or standard mean difference and the confidence interval for those on all the aforementioned medications have a considerable overlap, failing to show any priority of one medication over another in SER or SET. Likewise, the subgroup analysis illustrating the influence of MET therapy on the SER or SET in children aged 2–7 years versus children aged 7–14 years revealed a similar risk ratio of SER or a difference in the means of SET for both groups, again failing to reveal any effect of children’s age on the SER or SET. The subgroup analysis investigating the influence of the stone size on the SER or SET failed to reveal observable differences in SER or SET based on the different sizes of ureteral stones (<5 mm vs. 5–10 mm). Three studies reported adverse effects which included nasal congestion and headache/dizziness. The subgroup analysis failed to reveal more frequency of one adverse effect over another in the studied reports, and there was considerable overlap of the confidence intervals for the adverse effects (Table 4). The influence of gender on the SER or SET was reported in only one study [12] The overall stone expulsion rate for girls was 77.8% (14/18) versus 76.2% (16/21) for boys, which was not statistically significant (*p* = 0.45). Likewise, the mean± SD of the stone expulsion time for girls versus boys in total (case + control) was 5.85 ± 1.80 vs. 6.1 ± 2.0 days, which was not statistically significant (*p* = 0.60).

## 4. Discussion

The stone-free rate after 4 weeks in the MET group was 1.42 times that of the control group, and the stone expulsion time decreased by an average of 5.18 days. These findings were independent of the type of medication, the stone size, and the age of patients. 

The incidence of pediatric urolithiasis is increasing globally and is endemic in certain developing countries including India, Turkey, Pakistan, and the Far East [17,18]. Several factors play a role in the development of nephrolithiasis in children, including diet, environment, urinary tract infections, anatomical anomalies, and metabolic disorders [19]. Urolithiasis treatment modalities have changed during the last 20 years from open surgery to ESWL, PCNL, ureterorenoscopy, and laparoscopic or robot-assessed ureteropyelolithotomy [20,21,22]. Although there are successful efficacy results of the aforementioned procedures, they are not risk-free and can lead to complications. In addition, having high costs and anesthesia risks makes these procedures not the first choice [23,24]. Additionally, watchful waiting and consuming pharmacological agents such as calcium channel blockers, alpha blockers, and corticosteroids are considered treatment options for ureteric stones [12]. Several factors are associated with the spontaneous expulsion of distal ureteral stones, including the size, number, location, and structure of the stone, ureteral edema, and smooth muscle spasms [25,26]. Using medical expulsive therapy as a stone treatment leads to both an increase in the stone expulsion rate and a decrease in the risk of infection, fever, and the need for surgical interventions [12]. 

Although the present study was not limited to the usage of alpha blockers as MET in pediatric populations, all RCTs were performed on the efficacy of alpha blockers. Since 2002, many studies have been published on the beneficial effect of alpha blockers in enhancing the spontaneous expulsion of especially distal ureter stones through decreasing the intramural pressure and muscle tone of the ureter [27,28]. Previous studies reported the usage of alpha antagonists in pediatric urological disorders including dysfunctional voiding [29], bladder neck dysfunction [30], neurogenic bladder, and idiopathic urethritis [12]. Due to the efficacy and safety of alpha blockers in previous studies, a few studies used alpha antagonists as MET for distal ureteral stones in pediatric patients. Alpha1-receptors, especially subtype D, are mainly located in the distal part of the ureter, and by the blockage of alpha1-adrenoceptors by its antagonists, ureteral contractions and intraureteral pressure are reduced, and this may lead to stone passage [31,32]. 

While a significant number of studies evaluating MET among adult populations have been performed, only a few studies were dedicated to pediatric patients. The current study reviewed six RCTs involving 415 participants with distal ureter stones with a maximum size up to 10 mm, except one study in which the maximum stone size was 12 mm. Two studies evaluated doxazosin 0.03 mg/kg daily [12,14], which had contrary results. In one study [12], a slight increase in SER and a decrease in SET and pain episodes were reported, which were not statistically significant (*p* > 0.05). However, another study [14] reported a significantly higher SER (*p* = 0.005), a shorter SET (*p* = 0.001), and fewer colic pain episodes (*p* = 0.023) in patients consuming doxazosin. Additionally, the latter study mentioned that the greater stone expulsion rate was among stones <5 mm (*p* = 0.007) and children aged <7 years (*p* = 0.009). Studies on the adult population who consumed doxazosin indicated similar results, including an earlier stone expulsion rate, fewer pain colic episodes, and the absence of side effects [33]. Only Erturhan et al. [14] reported one child suffering from side effects (somnolence, nausea, and vomiting) and discontinued treatment. In considering the discrepancy of the results between the two above-mentioned studies, one should note that a statistically significant difference in the mean size of ureteral stones was reported between the case and the control group in the study of Aydogdou et al., who failed to demonstrate a statistically significant improvement in SER or SET through MET (case group stone size: 7.1 ± 1.3 mm versus control group stone size 5.8 ± 0.9 mm; *p* < 0.05) [12]. The pretreatment difference in stone size might adversely affect the findings of this study.

Three studies [3,15,16] assessed the efficacy of tamsulosin regarding the passage of ureteral stones in children. Mokhless et al. [15] reported successful outcomes including a significantly higher SER (*p* < 0.01), a shorter SET (*p* < 0.001), fewer pain episodes (*p* < 0.02), and less of an analgesic need (*p* < 0.02). In this study 14 and 1 patients out of 33 patients in the case group had distal ureteric stone fragments which were produced after ESWL and PCNL, respectively. The two other studies [3,16] reported similar findings. Three RCTs [3,15,16] reported minor side effects including orthostatic hypotension, somnolence, dizziness, headache, nausea, and nasal congestion, and none of them led to the withdrawal of any patient. A retrospective study by Tasian et al. [34] reported similar findings in favor of the spontaneous expulsion rate of ureteric stones, while the authors reported no tamsulosin-related side effect. In contrast, in the study conducted by George et al. [35], there were no significant differences in the need for surgical procedures and the number of hospital visits among children who used MET versus control children. 

The largest RCT, which enrolled 167 children to evaluate the efficacy of tamsulosin and silodosin, concluded that silodosin provides significantly better outcomes, including a higher stone expulsion rate (*p* = 0.04) and a shorter stone expulsion time (*p* < 0.001), than tamsulosin [16]. Fewer colic pain episodes were reported in patients consuming silodosin compared to those consuming tamsulosin; however, the difference was not statistically significant (*p* = 0.8). Regarding the safety issue, adverse effects, including dizziness, orthostatic hypotension, headache, nasal congestion, and nausea (which were grade 1 according to the Common Terminology Criteria for Adverse Events Ver. 4.0), were reported. On the contrary, Elgalaly et al. [13] disclosed no significant differences regarding the stone-free rate at 2 weeks (*p* = 0.4) and 4 weeks (*p* = 0.4). However, significantly shorter stone expulsion times (*p* = 0.02) and fewer pain episodes were reported among patients who were prescribed silodosin (*p* < 0.001). Only three patients (16.7%) consuming silodosin had mild headaches and dizziness. In the adult population given silodosin, a significantly greater stone expulsion rate for distal ureter stones was reported; however, this observation failed to happen for proximal and mid ureteral stones [36]. The higher stone expulsion rate and improved stone expulsion time reported by Soliman et al. [16] for silodosin in comparison with tamsulosin have also been reported in adult patients [37]. In all of the investigated RCTs, the treatment and control groups were prescribed ibuprofen for the control of renal colic pain. Analgesic medication constitutes an important part of the management protocol for renal colic during MET, giving the patient adequate pain control. Two major categories of analgesic medication are available for pediatric patients, namely, ibuprofen and acetaminophen. The major criteria that should be considered for the choice of an analgesic medication are pain control versus adverse effects. Regarding efficacy, in adult patients, NSAIDs were the primary choice of medication for pain control [38]. In pediatric patients, some authors demonstrated a better efficacy of ibuprofen versus acetaminophen for the treatment of fever and pain [39]. As stated above, safety is another important consideration. The published literature reveals a comparable adverse effect profile of short-term ibuprofen usage in comparison with acetaminophen in pediatric patients [39,40,41,42]. However, some articles point to a higher incidence of ibuprofen-induced adverse effects when used in younger children (<2 years or <6 months), and care should be taken to prescribe ibuprofen for young children. Likewise, ibuprofen may not be a good analgesic medication for children with asthma or wheezing, diarrhea, vomiting or dehydration, or preexisting kidney malfunction [43].

The current meta-analysis suffers from the following limitations: The total number of included RCTs is still relatively limited. While the sample size of the current meta-analysis is higher than those of earlier studies, a larger number of cases treated is needed to perform a powerful subgroup analysis on the factors influencing SER or SET. All of the investigated RCTs were performed in only two countries. The authors of these studies failed to provide detailed data on the SERs or SETs of their studies based on different categories of age, stone size, or medications. 

## 5. Conclusions

The present analysis strengthens the value of alpha-adrenergic blockers in improving the stone expulsion rate and stone expulsion time in children aged 2–14 years. The side effects associated with these medications were minor (headache, dizziness, or nasal congestion) and did not result in the discontinuation of therapy in most patients. The results of this meta-analysis failed to indicate any superiority of any studied alpha-adrenergic blockers, namely, silodosin, tamsulosin, and doxazosin, with respect to the stone expulsion time, rate, or adverse effects. The efficacy of medical expulsion therapy was not dependent on the age of children (2–6 years versus 7–14 years) nor on the size of ureteral stones (<5 mm versus 5–10 mm).

## Figures and Tables

**Figure 1 jcm-12-01410-f001:**
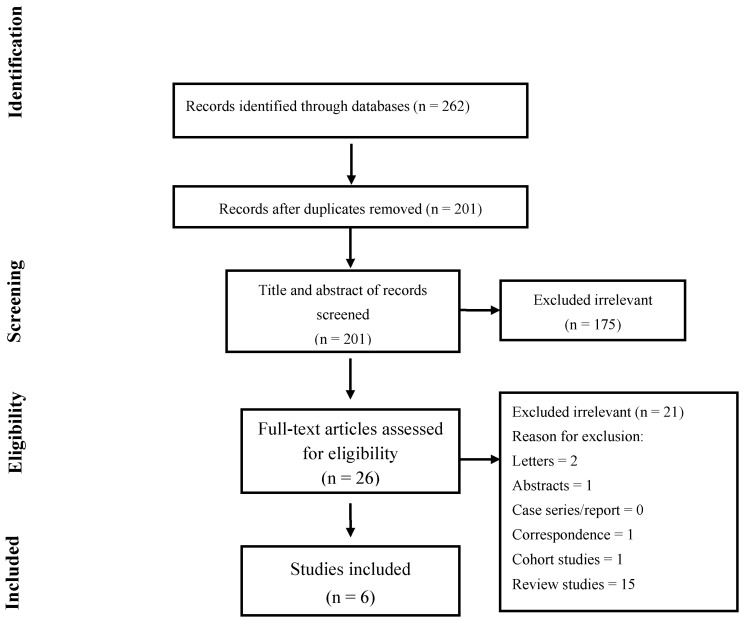
Flow chart of the study selection for inclusion in the systematic review.

**Figure 2 jcm-12-01410-f002:**
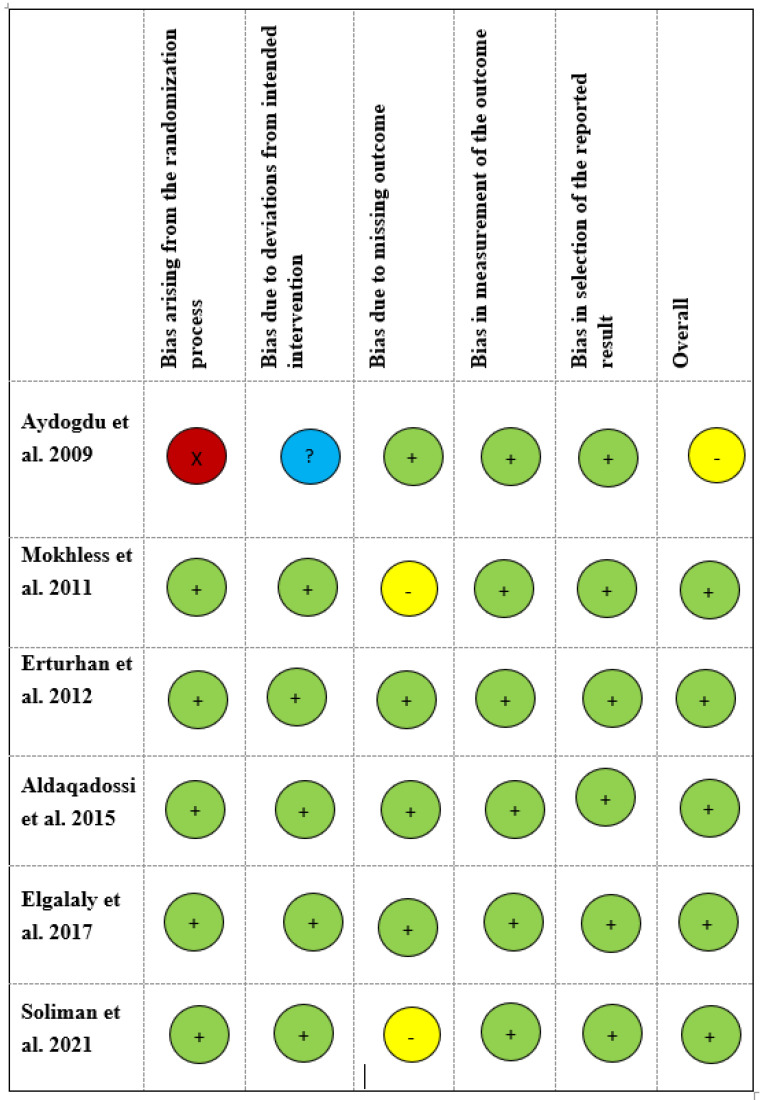
Risk of bias in included studies. A green circle indicates a low, a yellow circle indicates a moderate, a red circle indicates a critical, and a blue circle indicates an unknown level of bias [3,12,13,14,15,16].

**Figure 3 jcm-12-01410-f003:**
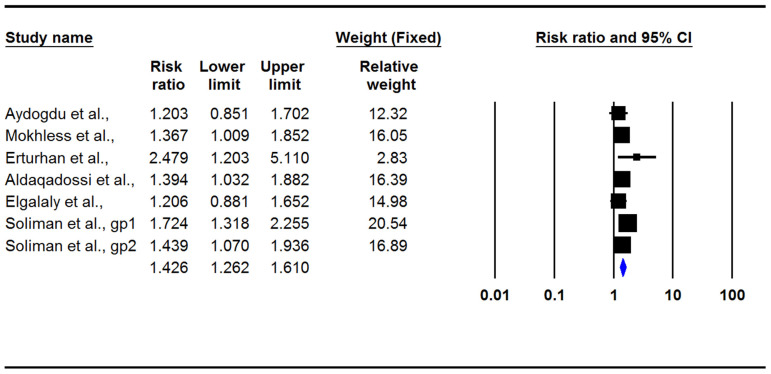
Forest plot for the risk ratios of the stone expulsion rate [3,12,13,14,15,16].

**Figure 4 jcm-12-01410-f004:**
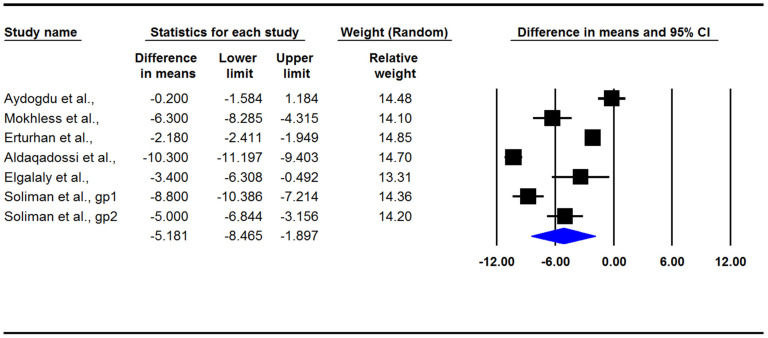
Forest plot for the differences in the means of the stone expulsion time [3,12,13,14,15,16].

**Figure 5 jcm-12-01410-f005:**
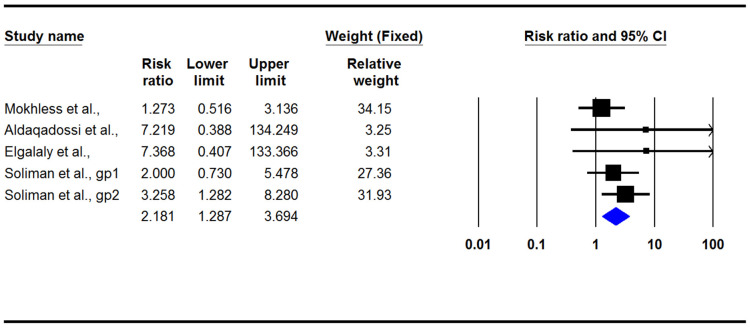
Forest plot for the risk ratios of adverse effects [3,13,15,16].

**Table 1 jcm-12-01410-t001:** Characteristics of the included studies.

Author	Country	Published Year	Gender (Boys/Girls)	Mean Patient Ages ± SD (yrs) Case/Control	Mean Stone Size ± SD (mm) Case/Control	Intervention/Control Treatment	Follow-up Duration (Weeks)
Aydogdu [12]	Turkey	2009	21/18	6.2 ± 2.4/5.1 ± 2.2	7.1 ± 1.3/5.8 ± 0.9	Doxazosin + Ibuprofen/Ibuprofen ^a^	3
Mokhless [15]	Egypt	2011	36/25	7.3 ± 4.2/7.1 ± 3.2	8.2 ± 2.3/7.8 ± 3.1	Tamsulosin + Ibuprofen/Ibuprofen+ Placebo ^b^	4
Erturhan [14]	Turkey	2012	24/21	6.0 ± 3.5/7.2 ± 3.5	4.58 ± 1.7/4.45 ± 1.5	Doxazosin + Ibuprofen/Ibuprofen ^a^	3
Aldaqadossi [3]	Egypt	2015	36/27	7.7 ±3.02/7.25 ± 2.70	6.52 ± 1.8/6.47 ±1.79	Tamsulosin + Ibuprofen/Ibuprofen ^c^	4
Elgalaly [13]	Egypt	2017	27/13	8.4 ± 3.1/7.7 ± 2.3	6.6 ± 1.7/6.7 ±1.4	Silodosin + Ibuprofen/Ibuprofen+ Placebo ^d^	4
Soliman [16]	Egypt	2021	107/60	Tamsulosin: 11.4 ± 2.4; Silodosin: 11.1 ± 2.8/11.2 ± 2.6	Tamsulosin: 6.3 ± 0.9; Silodosin: 6.2 ± 1.2/6.5 ± 1	Tamsulosin or Silodosin + Ibuprofen/Ibuprofen + Placebo ^e^	4

^a^ Doxazosin 0.03 mg/kg/d at bedtime. Ibuprofen 20 mg/kg/d divided in two equal doses. ^b^ Tamsulosin was 0.4 mg/d for >4 years and 0.2 mg/d for <4 years at bedtime. Ibuprofen given as the standard dose. ^c^ Tamsulosin was 0.4 mg/d for >5 years and 0.2 mg/d for ≤5 years. Ibuprofen 4–10 mg/kg every 6–8 h, as needed. ^d^ Silodosin 4 mg at bedtime. Ibuprofen 20 mg/kg/d in two equal doses. ^e^ Tamsulosin 0.4 mg/d. Silodosin 4 mg/d. Ibuprofen 4–10 mg/kg, as needed.

**Table 2 jcm-12-01410-t002:** Pooled risk ratios for the stone expulsion rate among subgroups of treatment regimens, sizes of stones, and ages of children.

Subgroups	No. of Studies	No. of Patients	Risk Ratio (95% CI) (*p*-Value)	Heterogeneity I^2^ (%)
Treatment regimens:				
Doxazosin-containing regimens	2 studies	84	1.6 (0.8–3.2) (0.18)	67.9
Tamsulosin-containing regimens	3 studies	235	1.4 (1.1–1.6) (0.000)	0.0
Silodosin-containing regimens	2 studies	149	1.4 (1.0–2.0) (0.035)	65.1
Size of stone:				
<5	3 studies	73	1.3 (1.0–1.6) (0.021)	34.4
5–10	3 studies	68	1.4 (0.9–2.1) (0.07)	6.2
Age:				
2–7	2 studies	49	1.5 (0.6–3.4) (0.32)	72.9
7–14	2 studies	35	1.4 (0.8–2.5) (0.20)	0.0

**Table 3 jcm-12-01410-t003:** Pooled differences in stone expulsion time among subgroups of treatment regimens and sizes of stones.

Subgroups	No. of Studies	No. of Patients	Std Diff in Means (95% CI) (*p*-Value)	Heterogeneity I^2^ (%)
Treatment regimens:				
Doxazosin-containing regimens	2 studies	84	−2.7 (−8.1/2.5) (0.30)	98.2
Tamsulosin-containing regimens	3 studies	235	−2.6 (−4.6/−0.7) (0.008)	96.6
Silodosin-containing regimens	2 studies	149	−1.4 (−2.7/−0.1) (0.028)	89.9
Size of stone:				
<5	3 studies	74	−1.4 (−3.0/0.1) (0.06)	87.7
5–10	3 studies	68	−2.9 (−6.1/0.2) (0.30)	94.7

**Table 4 jcm-12-01410-t004:** Pooled risk ratios for subgroups of adverse events.

Adverse Event	No. of Studies	No. of Patients	Risk Ratio (95% CI) (*p*-Value)	Heterogeneity I^2^ (%)
Nasal congestion	3 studies	291	1.7 (0.6–5.2) (0.28)	0.0
Headache and/or dizziness	3 studies	265	2.4 (0.8–6.8) (0.08)	0.0

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
