# Peer review of "Medical Expulsive Therapy for Pediatric Ureteral Stones: A Meta-Analysis of Randomized Clinical Trials"

_jcm, 2023, doi:10.3390/jcm12041410_

Round 1

Reviewer 1 Report

Abstract : the conclusion is not adequate : MET is associated with higher SFR and SET but higher adverse events. The authors can’t state it is safe.

Please modify

Introduction :

Adequate, the authors emphasize the absence of novelty of the topic and that their work is an update meta-analysis.

Methods :

The methodology is overall satisfying. The inclusion criterias are well defined and adequate for the topic of this study. The risk of bias and statistical analysis is ok.

Results : this section is well presented; the figures and tables are clear. The authors has to be congratulated for that.

Discussion : the discussion contains a section about the risk of prescription in children, which is mandatory. This aspect should also be in the conclusion. The authors should comment on the ibuprofen use in many RCT in children. Is it safe? Adverse events? The alphablockers are identically not the same in all RCT, what should be the influence of that regarding the data for adults. To continue the parallel with adults, the size of the stone has to be compared between adults (5-10mm, distal ureter) and children (all stones)

Conclusion : the conclusion is adequate but the risk of prescription in children has to be specifically mentioned to my opinion

References :

I would add a recommendation paper for ureteral stone in children. If there is not I would cite the current reco paper also

Author Response

Dear Editors of JCM

We would like to express our gratitude for reviewing our manuscript and providing valuable constructive comments. Below we have provided a point to point answer to the comments of the reviewers. We hope that the following responses will fulfill the expectations of the respected reviewers and the editorial team and that the revised file would be satisfactory for publication.

Regards,

Dr. Amir H Kashi

The Corresponding author

Reviewer 1:

COMMENT: Abstract: the conclusion is not adequate: MET is associated with higher SFR and SET but higher adverse events. The authors can’t state it is safe. Please modify

ANSWER: The abstract conclusion was revised and the following sentence was added as I quote: “They increase the stone expulsion rate and decrease the stone expulsion time; however, at a higher rate of adverse effects which include headache, dizziness, or nasal congestion.”

COMMENT: Introduction: Adequate, the authors emphasize the absence of novelty of the topic and that their work is an update meta-analysis.

ANSWER: We thank the respected reviewer for finding the submission satisfying.

COMMENT: Methods: The methodology is overall satisfying. The inclusion criterias are well defined and adequate for the topic of this study. The risk of bias and statistical analysis is ok.

ANSWER: We thank the respected reviewer for finding the submission satisfying.

COMMENT: Results: this section is well presented; the figures and tables are clear. The authors has to be congratulated for that.

ANSWER: We thank the respected reviewer for finding the submission satisfying.

COMMENT: Discussion: the discussion contains a section about the risk of prescription in children, which is mandatory. This aspect should also be in the conclusion. The authors should comment on the ibuprofen use in many RCT in children. Is it safe? Adverse events? The alphablockers are identically not the same in all RCT, what should be the influence of that regarding the data for adults. To continue the parallel with adults, the size of the stone has to be compared between adults (5-10mm, distal ureter) and children (all stones)

ANSWER: We included the following section to the discussion section to highlight concern relatef to the use of ibuprofen in children as I quote:” In all investigated RCTs, the treatment and control groups were prescribed ibuprofen for control of renal colic pain. Analgesic medication constitutes an important part of the management protocol for renal colic during MET to give the patient adequate pain control. Two major categories of analgesic medication are available for pediatric patients namely ibuprofen and acetaminophen. The major criteria which should be considered for the choice of an analgesic medication are pain control versus adverse effects. Regarding efficacy, in adult patients, NSAIDs were the primary choice of medication for pain control (Türk et al.). In Pediatric patients, some authors demonstrated better efficacy of ibuprofen versus acetaminophen for the treatment of fever and pain (Pierce et al.) As stated above, safety is another important consideration. The published literature reveals a comparable adverse effect profile of short term ibuprofen usage in comparison with acetaminophen in pediatric patients (Pierce et al., Lesko et al., Walsh et al., Ashraf et al.). However, some articles point to a higher incidence of ibuprofen-induced adverse effects when used in younger children (<2 years or < 6 months) and care should be taken to prescribe ibuprofen for young children. Likewise, ibuprofen may not be a good analgesic medication for children with asthma or wheezing, diarrhea, vomiting or dehydration, or preexisting kidney malfunction (de Martino et al.)”

Regarding comparison of alpha-blockers, our submission failed to find any superiority of one alpha-blocker over another as already stated in page 8 line 165-166.

Comparison of stone size did not disclose any influence of stone size in the outcome of MET as already mentioned in page 8 line 169-171.

COMMENT: Conclusion: the conclusion is adequate but the risk of prescription in children has to be specifically mentioned to my opinion

ANSWER: We revised the conclusion to more evidently include reference to adverse effects in line 276-278, as I quote: “The side effects associated with medications are minor (headache, dizziness, or nasal congestion) and did not result in the discontinuation of therapy in most patients.”

COMMENT: References: I would add a recommendation paper for ureteral stone in children. If there is not, I would cite the current reco paper also

ANSWER: We included a reference from the EAU guideline urolithiasis editors on the use of MET authored by Turk et al. entitled: “EAU Guidelines on Diagnosis and Conservative Management of Urolithiasis. Eur Urol. 2016;69(3):468-74.”

Reviewer 2 Report

1)   General comments

The authors aim to evaluate the efficacy and safety of medical expulsive therapy (MET) for ureteral stones in pediatric patients. This systematic review showed the stone free rate after 4 weeks in the MET group was 1.42 times the control group and the stone expulsion time decreased by an average of 5.18 days. These findings were independent of the type of medication, stone size, and age of patients. Alpha-blockers as medical expulsive therapy among pediatric patients are efficient and safe.

The reviewer generally agrees with the conclusion.

However, there are several issues need to improve. The reviewer would like suggests several issues as follows;

 2)   Specific comments for revision

a)   Minor

#1 What is the dose of each medication used?

#2 How long is the medication used?

#3 Are there any reports of a difference in stone expulsion by sexes?

Author Response

Dear Editors of JCM

We would like to express our gratitude for reviewing our manuscript and providing valuable constructive comments. Below we have provided a point to point answer to the comments of the reviewers. We hope that the following responses will fulfill the expectations of the respected reviewers and the editorial team and that the revised file would be satisfactory for publication.

Regards,

Dr. Amir H Kashi

The Corresponding author

Reviewer 2:

The reviewer generally agrees with the conclusion.

We thank the respected reviewer for finding the submission satisfying.

 COMMENT: What is the dose of each medication used?

ANSWER: The dose of medications was included in the footnote of Table 1.

COMMENT:  How long is the medication used?

ANSWER: The duration of medication was explained in a separate column in Table 1.

COMMENT: Are there any reports of a difference in stone expulsion by sexes?

ANSWER: Only the report by Aydogdu et al. evaluated the SER and SET divided by gender types. The comparison of SER by gender was not statistically significant as the average stone expulsion rate for girls was 77.8% versus 76.2% for boys. Likewise, the comparison of the stone expulsion time between girls and boys in each treatment group (case and control) and in total was not statistically significant. We explained on this title in result section as I quote: “The influence of gender on SER or SET was reported only in one study (Aydogdu et al.) The overall stone expulsion rate for girls was 77.8% (14/18) versus 76.2% (16/21) for boys which was not statistically significant (P=0.45). Likewise, the mean± SD of stone expulsion time for girls versus boys in total (case+ control) was 5.85±1.80 vs 6.1±2.0 which was neither statistically significant (P=0.60).”